# How South Africa Used National Cycle Threshold (Ct) Values to Continuously Monitor SARS-CoV-2 Laboratory Test Quality

**DOI:** 10.3390/diagnostics13152554

**Published:** 2023-08-01

**Authors:** Lesley Erica Scott, Nei-yuan Hsiao, Graeme Dor, Lucia Hans, Puleng Marokane, Manuel Pedro da Silva, Wolfgang Preiser, Helena Vreede, Jonathan Tsoka, Koleka Mlisana, Wendy Susan Stevens

**Affiliations:** 1Wits Diagnostic Innovation Hub, Faculty of Health Science, University of the Witwatersrand, Johannesburg 2093, South Africalucia.hans@nhls.ac.za (L.H.); pedro.dasilva@nhls.ac.za (M.P.d.S.); jtsoka@ilead.org.za (J.T.); wendy.stevens@wits.ac.za (W.S.S.); 2Division of Medical Virology, Faculty of Heath Sciences, University of Cape Town, Cape Town 7700, South Africa; marvin.hsiao@uct.ac.za; 3The National Health Laboratory Service, Johannesburg, Private Bag X8, Sandringham 2131, South Africa; preiser@sun.ac.za (W.P.); helena.vreede@nhls.ac.za (H.V.); koleka.mlisana@nhls.ac.za (K.M.); 4The National Priority Program of the National Health Laboratory Service, Johannesburg, Private Bag X8, Sandringham 2131, South Africa; puleng.marokane@nhls.ac.za; 5Department of Pathology, Faculty of Medicine and Health Sciences, Stellenbosch University, Stellenbosch 7600, South Africa; 6Division of Chemical Pathology, Faculty of Heath Sciences, University of Cape Town, Cape Town 7700, South Africa

**Keywords:** SARS-CoV-2, COVID-19, laboratory diagnostics, cycle threshold, PCR platforms, median Ct, moving average, variants, continuous quality monitoring

## Abstract

The high demand for SARS-CoV-2 tests but limited supply to South African laboratories early in the COVID-19 pandemic resulted in a heterogenous diagnostic footprint of open and closed molecular testing platforms being implemented. Ongoing monitoring of the performance of these multiple and varied systems required novel approaches, especially during the circulation of variants. The National Health Laboratory Service centrally collected cycle threshold (Ct) values from 1,497,669 test results reported from 6 commonly used PCR assays in 36 months, and visually monitored changes in their median Ct within a 28-day centered moving average for each assays’ gene targets. This continuous quality monitoring rapidly identified delayed hybridization of *RdRp* in the Allplex™ SARS-CoV-2 assay due to the Delta (B.1.617.2) variant; *S*-gene target failure in the TaqPath™ COVID-19 assay due to B.1.1.7 (Alpha) and the B.1.1.529 (Omicron); and recently *E*-gene delayed hybridization in the Xpert^®^ Xpress SARS-CoV-2 due to XBB.1.5. This near “real-time” monitoring helped inform the need for sequencing and the importance of multiplex molecular nucleic acid amplification technology designs used in diagnostics for patient care. This continuous quality monitoring approach at the granularity of Ct values should be included in ongoing surveillance and with application to other disease use cases that rely on molecular diagnostics.

## 1. Introduction

South Africa (population of ~60.6 million people) [1] has one of the highest HIV and TB prevalence rates in the world: 8.45 million people live with HIV (13.9% of the population) [1] and a TB incident rate of 513/100,000 is reported [2]. Healthcare is funded through the government for 84% of the population (public sector) and the remainder is funded privately through individuals, medical schemes, and insurance companies [3]. Laboratory services are provided through networks of private and public sector laboratories, with the latter comprising 256 National Health Laboratory Service (NHLS) facilities positioned centrally and at district level across the nine provinces.

On 5 March 2020, South Africa reported their first case of COVID-19 [4] and both public and private laboratories’ [5] virology services had to rapidly scale their SARS-CoV-2 nucleic acid amplification technology (NAAT); the primary method of diagnosing infection with SARS-CoV-2 [6,7]. Within one month, however, the country’s SARS-CoV-2 testing demands required expanding molecular testing to the national priority program (NPP) of the NHLS. The NPP has capacity for 10 million molecular tests/annum, which predominantly support HIV viral load (VL) monitoring [8] and TB molecular diagnostics [9]. Testing is performed in 17 VL laboratories (11 also provide HIV early infant diagnostics) equipped with cobas^®^ [Roche Molecular, Pleasanton, CA, USA] and Alinity m [Abbott Molecular, Des Plains, IL, USA] platforms) and 175 GeneXpert (Cepheid, Sunnyvale, CA, USA) testing laboratories, which service ~3800 primary health care facilities. Test kit demands for the NPP’s closed platforms (until 2021), however, could not be met due to the inability of suppliers to ship to South Africa as demands escalated in their countries of manufacture. SARS-CoV-2 testing therefore had to expand to a third network of laboratories. This involved a SARS-CoV-2 surge program initiated by the South African Medical Research Council (SAMRC) [10] with solidarity funding to support selected academic laboratories, councils, and research institutes to conduct SARS-CoV-2 testing. A rapid laboratory electronic assessment tool was developed to collect information on site location, ISO 15189 status, Health Professions Council of South Africa status of key staff, SARS-CoV-2 readiness (available testing platforms) and implemented quality management systems (trained staff, use of electronic requisition forms, use of barcodes, PPE stock, kit and supply storage and procurement processes, laboratory information (LIS) and IT support on-site). Thirty-two sites were assessed by the NHLS’ Quality Assurance Division and eleven were selected for SARS-CoV-2 surge testing. Testing in these laboratories commenced in June 2020 and continued until January 2021.

Test results (SARS-CoV-2 detected, SARS-CoV-2 not detected and test unsuccessful) from the private laboratories, the surge testing sites and the NHLS (including NPP) were reported as cases by the National Institute of Communicable Diseases (NICD), who provide central COVID-19 epidemiology and surveillance reporting nationally [11] and regionally [12]. Overall, the private laboratories performed ~55% of South Africa’s SARS-CoV-2 diagnostics (as reported in week 34 of 2021 [13]). A link was also established to the genomics network with five laboratories identified across South Africa capable of performing SARS-CoV-2 sequencing [14]. Throughout the five COVID-19 infectious waves in South Africa [15], specimens were shared between the diagnostic laboratories and genomics group to detect SARS-CoV-2 variants, including variants of concern (VOC).

Laboratories are required to participate in external proficiency testing to monitor pre-analytical, analytical and post-analytical performance. However, after decades of quality indicators, a paradigm shift should be towards indicators of total quality [16]. Figure 1 briefly outlines some of the processes required for quality management prior to selecting suitable laboratory tests, implementing tests in the field and then data environments for result reporting. During large scale pandemic testing, such as during COVID-19, external quality assessment programs were implemented [17,18,19]. However, these would not easily identify quality issues timeously. The South African NHLS approached this challenge by including the cycle threshold (Ct) values for each gene target of the SARS-CoV-2 tests performed in their laboratories that identified the presence of SARS-CoV-2 (and therefore diagnostic for COVID-19) in their LIS interface. The NHLS has a single national LIS, TrakCare (InterSystems, Cambridge, MA, USA), where all SARS-CoV-2 diagnostic systems are interfaced, and Ct values accessed for analysis from a central data warehouse (Netezza, IBM, USA-based server in Johannesburg).

A Ct value is intrinsic to PCR assays and is a measure of the amount of target nucleic acid in the specimen [20]. SARS-CoV-2 test manufacturers provide information on which SARS-CoV-2 gene regions their assays target and provide the Ct values in the test output comma separator value (.csv) files. Therefore, in addition to central monitoring of daily test volumes, qualitative test results and indicators to track changes in South Africa’s COVID-19 epidemic, the NHLS’ centrally collected SARS-CoV-2 Ct values were analyzed.

The aim of this study was therefore to highlight how the Ct values from the ensemble of closed and open testing platforms and assays in the NHLS laboratories were used for continuous quality monitoring (CQM) of diagnostic assay performance.

## 2. Materials and Methods

Specimens received in the NHLS laboratories for SARS-CoV-2 testing were collected using nasopharyngeal and oropharyngeal swabs. These were transported dry during shortages of universal transport medium (UTM) and were cut and placed into phosphate buffered saline upon arrival and processed according to standard operating procedures. Specimens were registered in the LIS and tested across 205 NHLS laboratories using the locally available platforms and testing protocols. Test results, including Ct values (when SARS-CoV-2 was detected) were accessed from the CDW through an extract, transform and load process to generate a .csv file for analysis. Data did not include patient unique identifiers, and hence analyses performed included longitudinal follow-up testing. Ethics approval was obtained from the University of the Witwatersrand Human Research Ethics Committee (number M160978), Johannesburg, South Africa. Data files were analyzed and data visualized using STATA 14 (StataCorp. 2015, Stata Statistical Software: Release 14. College Station, TX, USA: StataCorp LP) and Tableau 2020.3 (Tableau Software. 2020. Seattle, WA, USA: Tableau). Bar charts depicted daily positive test numbers and line graphs represented changes in median Ct values within a 28-day centered moving average. Changes in patterns of the moving average of the median Ct values of one or more gene targets could reflect an assay performance issue (quality or even lot variability) or due to the introduction of a SARS-CoV-2 mutation that impacts PCR efficiency. The latter could lead to primer/probe reduced hybridization (where the median Ct value would increase above other unaffected targets) or complete drop-out (where the median Ct value would decrease substantially due to the recording of zero Ct values in contrast to unaffected target Ct values). These changes were monitored across all assays and their gene targets, and no Ct values were modified or removed to ensure consistent daily monitoring of raw data outputs that could inform critical alerts in near-real time. Ten SARS-CoV-2 molecular assays were recorded in the CDW database. However, 94.2% of tests were performed on six assays, which form the focus of this analysis. These assays are classified into open laboratory testing platforms: Allplex™ SARS-CoV-2 (SeeGene Inc., Seoul, Korea); TaqPath™ COVID-19 (Thermo Fisher Scientific, Waltham, MA, USA) and closed platforms: cobas^®^ SARS-CoV-2 (Roche Molecular, Pleasanton, CA, USA); Xpert^®^ Xpress SARS-CoV-2 (Cepheid, Sunnyvale, CA, USA), RealTime SARS-CoV-2 (Abbott Molecular, Des Plains, IL, USA) and Alinity m (Abbott Molecular, Des Plains, IL, USA). Table 1 details the assay gene targets and range in Ct values when a specimen is reported positive for the detection of SARS-CoV-2. All assays report their gene targets in different fluorescent channels and can individually be identified, except the RealTime SARS-CoV-2 and the Alinity m, where both gene targets are reported in a single fluorescent channel and neither target can be individually identified.

## 3. Results

The six SARS-CoV-2 assays utilized by the NHLS were implemented at different times during the pandemic and their uptake at a province level varied as outlined in Figure 2. Across 36 months of testing (March 2020–March 2023), the Allplex™ SARS-CoV-2 and the Xpert^®^ Xpress SARS-CoV-2 contributed 48% of the test results. The Xpert^®^ Xpress SARS-CoV-2 assay is the only testing platform used in all provinces, which is a consequence of the placement of this platform by the NPP for use in the national TB diagnostic program. In contrast, the TaqPath™ COVID-19 assay was predominantly only used in the Gauteng Province. The RealTime SARS-CoV-2 and Alinity m assays only contributed ~10% towards testing, despite these platforms used for the HIV diagnostics program of NPP. At least 66% of all testing was reported from the three most densely populated provinces (Gauteng, KwaZulu-Natal, and the Western Cape).

A total of 8,573,872 tests were performed by NHLS laboratories during the study time period and 1,572,098 (18.33%) of them reported the presence of SARS-CoV-2, and 1,497,669 (95.2%) of these included Ct values for their gene targets. Ct values would not have been captured where pathology results were entered manually by some academic partners whose systems were not interfaced with the NHLS LIS. The daily median Ct for each SARS-CoV-2 gene target for the open and closed platforms is described in the following sections. Common to all assays (and visualized across all plots), is the decrease in median Ct (increase in SARS-CoV-2 viral concentration) during an infection wave and vice versa after an infection wave.

### 3.1. The NHLS SARS-CoV-2 Open Testing Platforms

#### 3.1.1. Allplex™ SARS-CoV-2 (SeeGene)

The Allplex™ SARS-CoV-2 assay was the first test to be implemented by NHLS in March 2020 and contributed a national test count of 2,095,588 million tests (26%) after 36 months. Although this assay was used by all provinces, 91% of the results were generated from the Gauteng, Western Cape, KwaZulu-Natal and Eastern Cape provinces. Figure 3 highlights that the *E*-gene generated the lowest Ct values followed by the *RdRp* and *N*-gene Ct values. The changes in 28-day centered moving average of the median Ct for all three genes mirror (duplicate) each other’s visual pattern except during South Africa’s third COVID-19 wave, which peaked in July 2021. The *RdRp* median Ct increased above the *N*-gene Ct. This increase in Ct (reduced PCR performance) was reported to be a result of the B.1.617.2 (Delta) variant [21] which had a highly conserved nonsynonymous mutation (G15451A) exclusively within the *RdRp* gene, and thereby negatively affecting the *RdRp* PCR efficiency of the Allplex™ SARS-CoV-2 assay.

This was continuously monitored by calculating the relative change in Ct between *RdRp* and *E* and any test result where the Ct of *RdRp-E* > 3.5 indicated the presence of the Delta variant. During the Delta wave, up to 62% of all positive specimens tested using the Allplex™ SARS-CoV-2 assay reported this phenomenon. No changes in this assays’ PCR efficiency were noted beyond the impact from the Delta variant. However, SeeGene introduced a new Allplex™ SARS-CoV-2 assay version that included the *S*-gene target, but this target is reported in the same PCR fluorescent channel as *RdRp* and therefore neither target’s PCR efficiency is discernable. Towards the end of the study period (March 2023), the Allplex™ SARS-CoV-2 testing volumes were reduced to ~42/day, making daily monitoring less reliable.

#### 3.1.2. TaqPath™ COVID-19 (Thermo Fisher Scientific)

The TaqPath™ COVID-19 assay was implemented in May 2020 and contributed to 1,660,419 million tests (21%) performed by the NHLS. Implementation, however, was not national, and 50% of testing was performed in the Gauteng Province. Figure 4 shows the Ct values of *ORF1ab*, *N* and *S*-genes generally mirrored each other for the first two waves. However, the S-gene pattern changed during the 3rd, 4th and 5th waves, with the median Ct of the *S*-gene target much lower than the *ORF1ab* and *N*-genes. This phenomenon was due to *S*-gene target failure (SGTF) [22], brought about by deletion of amino acids 69 and 70 in B.1.1.7 (Alpha) and B.1.1.529 (Omicron) spike genes. This yielded a distinct absent *S*-gene and hence no amplification during PCR and Ct values reported as zero in the TaqPath™ COVID-19 .csv file, which contributed to an overall low median *S*-gene Ct value. The TaqPath™ COVID-19 testing rates decreased to <4 tests/day at the end of March 2023, making continuous quality monitoring less reliable.

### 3.2. The NHLS SARS-CoV-2 Closed Testing Platforms

#### 3.2.1. Xpert^®^ Xpress SARS-CoV-2 (Cepheid)

The Xpert^®^ Xpress SARS-CoV-2 assay was implemented by the NHLS in March 2020 and contributed to 2,073,844 million tests (26%) after 36 months. This assay was used in all provinces across South Africa, however, Gauteng (21%), KwaZulu-Natal (17%), and the Western Cape (16%) accounted for more than half (54%) of the total Xpert^®^ Xpress SARS-CoV-2 test results. Figure 5 highlights a change in the NHLS’ testing algorithm implemented by their Xpert^®^ Xpress SARS-CoV-2 testing sites on 15 September 2021. Concerns were raised by the NHLS virology expert committee on reporting a positive SARS-CoV-2 result when Ct values of either or both gene targets approached the threshold of 45. The following algorithm was therefore implemented within the NHLS LIS: where a single E or N2 has a Ct > 38 or where both E and N2 Ct > 40, these specimens are reported as “inconclusive” (internal memo Dr M.P. da Silva). This change is evident in Figure 5 with greater mirroring (less variability in the Ct trends) between both gene targets beyond this date. Figure 5 further shows that the *E*-gene generated lower Ct values than the *N2*-gene for 33 of the 36 months. Changes in the 28-day centered moving average of the median Ct for both the *E* and *N2*-genes mirror each other with the exception of the period after January 2023, where the median *E*-gene Ct values increase. This phenomenon is due to XBB.1.5. VOC, which affects the *E*-gene coverage dropping by 1% due to two mismatches (personal communication from Cepheid medical affairs) that delay PCR hybridization. As the Xpert^®^ Xpress SARS-CoV-2 continues to be used in NHLS testing sites (400 tests/day at the end of March 2023), the proportion of specimens with *E*-gene Ct > *N2*-gene Ct can therefore be used to monitor the circulation of XBB.1.5. This was evident among 92% (28-day moving average) of Xpert^®^ Xpress SARS-CoV-2 test results at the end of March 2023.

#### 3.2.2. cobas^®^ SARS-CoV-2 (Roche Molecular)

The cobas^®^ SARS-CoV-2 assay was implemented in April 2020 and contributed 1,357,991 million tests (17%) after 36 months to the NHLS’ test volumes. Although the assay was implemented across all provinces in South Africa; the Eastern Cape (22%), KwaZulu-Natal (21%), and the Western Cape (15%) contributed 60% of the total testing volumes. Figure 6 clearly shows that changes in the 28-day centered moving average of the median Ct is perfectly aligned for both gene targets (*E* and *ORF1ab*). Their mirrored pattern remains throughout the 36 months of testing and this assay appears unaffected by any VOCs. SARS-CoV-2 testing on this platform however ceased in the NHLS in July 2022.

#### 3.2.3. RealTime SARS-CoV-2 and ALINITY m SARS-CoV-2 (Abbott Molecular)

The RealTime SARS-CoV-2 and ALINITY m SARS-CoV-2 assays, from a single supplier, were implemented by the NHLS in May 2020 and November 2020 respectively. The RealTime SARS-CoV-2 assay contributed 409,654 tests (5%) while the ALINITY m SARS-CoV-2 assay contributed 479,536 tests (6%) to the NHLS’ national testing volumes during the 36 months. A significant portion (62% of RealTime SARS-CoV-2 tests and 81% of ALINITY m SARS-CoV-2) of these tests were conducted by two provinces (KwaZulu-Natal and Gauteng). Both these assays target the *RdRP* and *N* gene regions, but neither region can be distinguished by a Ct value as both targets fluoresce in a single channel as outlined in Figure 7. SARS-CoV-2 testing continues to be reported on the ALINITY m SARS-CoV-2 platform (65 tests/day at the end of March 2023), however, testing was discontinued on the RealTime SARS-CoV-2 in July 2022.

## 4. Discussion

The Ct is a variable that correlates with the amount of target RNA in a specimen [23], and during COVID-19 the SARS-CoV-2 viral load in the respiratory tract was reported to align with an individual’s disease progression [6,24,25,26]. In this analysis, we showed that continuous quality monitoring of median Ct’s of SARS-CoV-2 gene targets at a national (population) level could rapidly identify changes in assay PCR gene target performance, due to genetic mutations of SARS-CoV-2 and assist in changing laboratory reporting algorithms. Our Ct analyses were therefore provided in weekly updates (in the form of written brief and weekly zoom meetings) to the South African scientific community involved in the COVID-19 responses. Changes in Ct values over time could therefore be an additional parameter included in good laboratory practices’ quality management systems, and especially within the context of gene mutations that could impact molecular assay performance.

Although the CQM of Ct values across the gene targets and across assays was limited to only the NHLS testing laboratories (<50% of South Africa’s SARS-CoV-2 testing), 1,497,669 Ct values were analyzed over 36 months. This “big” data also reduced the impact of variables (central limit theorem) such as specimen quality, PCR inhibitory substances, variable transport media, variable front-end extraction or extraction-free technologies and NAAT reaction volume, to name a few, that is reported as complexities in SARS-CoV-2 laboratory testing [27].Quality control practices at a laboratory level (EQA or inter-laboratory exercises) to assess the intrinsic laboratory quality and capacity are still necessary to be implemented.

Our findings show that of the six SARS-CoV-2 molecular assays implemented at scale across the NHLS, the Ct value analysis by gene target rapidly identified three assays’ targets affected by VOCs: (i) the B.1.617.2 (Delta) variant affected the Allplex™ SARS-CoV-2 assay in the *RdRp* target region [21]; (ii) the B.1.1.7 (Alpha) and the B.1.1.529 (Omicron) affected the TaqPath™ COVID-19 assay in the *S*-gene target region [15]; and (iii) the XBB.1.5 affected the Xpert^®^ Xpress SARS-CoV-2 in the *E*-gene target region. The cobas^®^ SARS-CoV-2 assay appeared not to be affected by circulating VOCs based on no changes identified in their gene target Ct value trends. The inability to distinguish the Ct values for the multiple gene targets in the RealTime SARS-CoV-2 and ALINITY m SARS-CoV-2 made the CQM less helpful in identifying changes in the assay quality potentially due to VOC but was a stable marker outlining the expected patterns in Ct change over the course of COVID-19 that other assays could be compared. Despite several assays’ gene target performance being affected by VOCs, their multiplex nature [28] (at least two viral genes targeted to increase the probability to identify the virus at low viral load and in the presence of viral mutations), enabled the assays to continue being used as the primary diagnostic for patient care [29,30].

Through the Ct CQM, the loss or change in target performance was, however, also quickly identified as an advantage in monitoring the spread of VOC [15,22] in “near” real-time with only a two-day lag period between specimen receipt and authorization. This also included monitoring the recovery of Ct values of affected gene targets, with the best example shown by the TaqPath™ COVID-19 where the B.1.1.7 (Alpha) caused the SGTF during wave 3 (which was, however, rapidly replaced by the B.1.617.2 (Delta) variant). Hence, the TaqPath™ COVID-19 *S*-gene target performance could inform changes in circulating variants.

Although the NHLS’ multi-assay implementation approach was governed by test demands and availability of platforms and reagents, it did prove possible to monitor such a multi-assay program through the unique centralized LIS. This in turn also highlighted some limitations, such as the TaqPath™ COVID-19 assay not being implemented nationally, and >50% of test results reported from only the Gauteng Province. Findings such as the SGTF therefore could not be extrapolated to regions where this assay was not being used.

Overall, this study is the first, to our knowledge, that highlights CQM using national program laboratory Ct values from multiple SARS-CoV-2 assays. The data strongly show that variables of molecular test results can be a key part of laboratory quality management. It also highlights the multidisciplinary approach to CQM with the need to understand molecular technology, the need to understand the role of diagnostics in clinical and laboratory practices and the need to understand big data analytics and visualization. This study also highlights the value molecular diagnostics “near-real-time” analysis has in informing the need for sequencing. The introduction of rapid SARS-CoV-2 antigen tests, and self-tests however severely limits this value and ongoing molecular surveillance should be maintained. This system’s approach to quality management and program performance monitoring therefore should also be investigated for other disease use cases, such as TB and HIV, where molecular technology is the primary diagnostic test.

## 5. Conclusions

In conclusion, this study demonstrates the potential of Continuous Quality Monitoring (CQM) using Ct values of SARS-CoV-2 gene targets at a national level to rapidly identify changes in SARS-CoV-2 laboratory assay performance, particularly in response to genetic mutations of the virus. Despite some limitations, such as the regional distribution of certain assays, the approach using centralized laboratory information emphasizes the significance of molecular diagnostics in informing clinical decisions and surveillance efforts but does require multidisciplinary collaboration between clinical, molecular and data scientists. Furthermore, the study suggests the potential application of this system’s approach to quality management in other disease contexts where molecular technology serves as a primary diagnostic tool. Overall, this research underscores the value of continuous monitoring in pandemic responses and the need to maintain molecular surveillance.

## Figures and Tables

**Figure 1 diagnostics-13-02554-f001:**
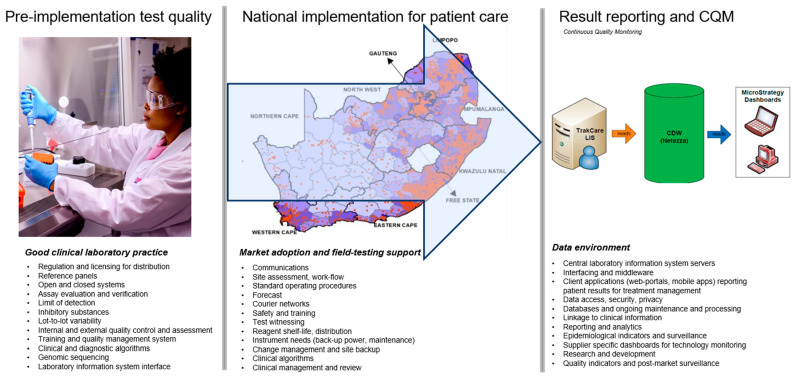
An overview of quality management systems required for laboratory diagnostics applied in patient care and the key role of laboratory information systems. This diagram highlights South Africa’s spread of care facilities (orange) across the nine provinces (listed on the South African map). The purple shading reflects population density at a district (municipality) level. The names of the nine provinces are listed on the map. Each instrument placed in the field within the NHLS is connected to a LIS (TrakCare), with data extracted, transformed and loaded and stored within a central data warehouse (CDW). This is then accessible for post-market surveillance and in the case of this study for continuous quality monitoring of the Ct values obtained from each test performed.

**Figure 2 diagnostics-13-02554-f002:**
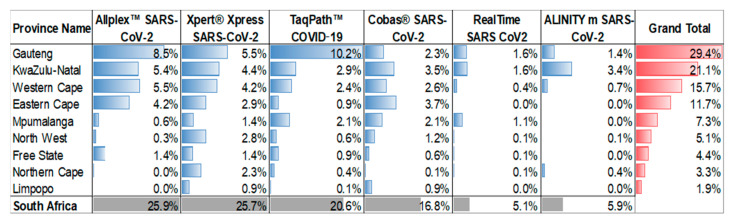
Proportion of molecular SARS-CoV-2 testing performed by NHLS laboratories between March 2020 and March 2023 across the six commonly used laboratory testing systems (accounting for 94.2% NHLS test results). The testing proportions are stratified by province and conditional formatting applied as colored bars.

**Figure 3 diagnostics-13-02554-f003:**
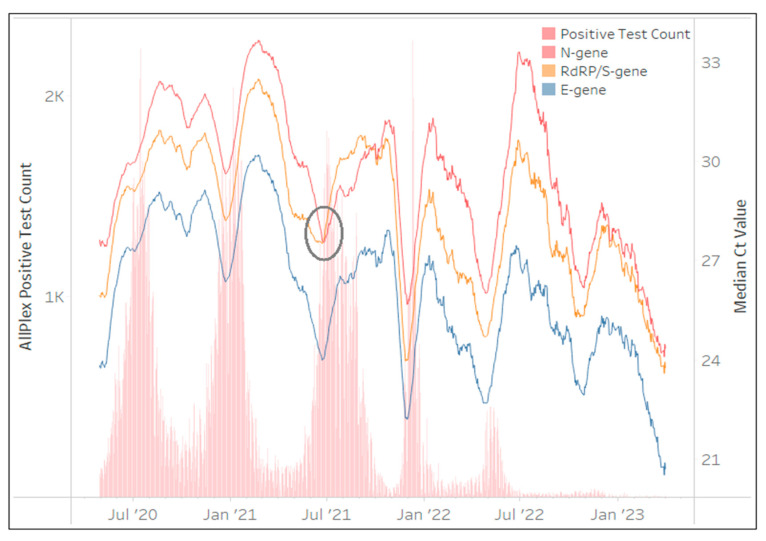
The number of Allplex™ SARS-CoV-2 tests (daily) reporting the presence of SARS-CoV-2 in red bars between March 2020 and March 2023 (primary vertical axis) overlaid with the median Ct values from each gene target in line plots. The median Ct value of the 28-day centered moving average for each gene target is represented on the secondary vertical axis. The key highlights the assay specific gene targets. A change in performance of the *RdRp*-gene target median Ct is visible over July 2021 (circled).

**Figure 4 diagnostics-13-02554-f004:**
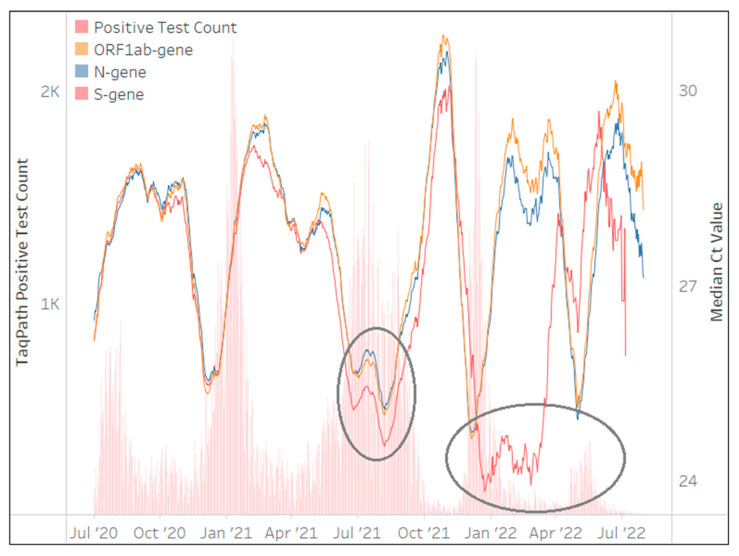
The number of TaqPath™ COVID-19 tests (daily) reporting the presence of SARS-CoV-2 in red bars between March 2020 and March 2023 (primary vertical axis) overlaid with the median Ct values from each gene target in line plots. The median Ct value of the 28-day centered moving average for each gene target is represented on the secondary vertical axis. The key highlights the assay specific gene targets. The impact of the B.1.1.7 (wave 3) and B.1.1.529 (waves 4 and 5) in causing the SFTG is evident by the *S*-gene median Ct diverging from the *ORF1ab* and *N*-gene’s Ct (circles).

**Figure 5 diagnostics-13-02554-f005:**
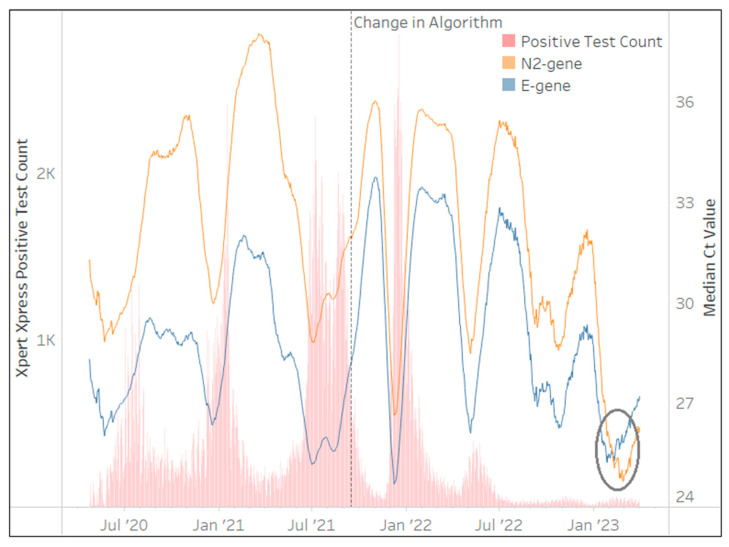
The number of Xpert^®^ Xpress SARS-CoV-2 tests (daily) reporting the presence of SARS-CoV-2 in red bars between March 2020 and March 2023 (primary vertical axis) overlaid with the median Ct values from each gene target in line plots. The median Ct value of the 28-day centered moving average for each gene target is represented on the secondary vertical axis. The key highlights the assay specific gene targets. A new specimen result reporting algorithm was implemented in September 2021 (dotted vertical line). The effect of the XBB.1.5. is evident in the last few months of 2023 (circle), which causes a delayed hybridization in the *E*-gene PCR.

**Figure 6 diagnostics-13-02554-f006:**
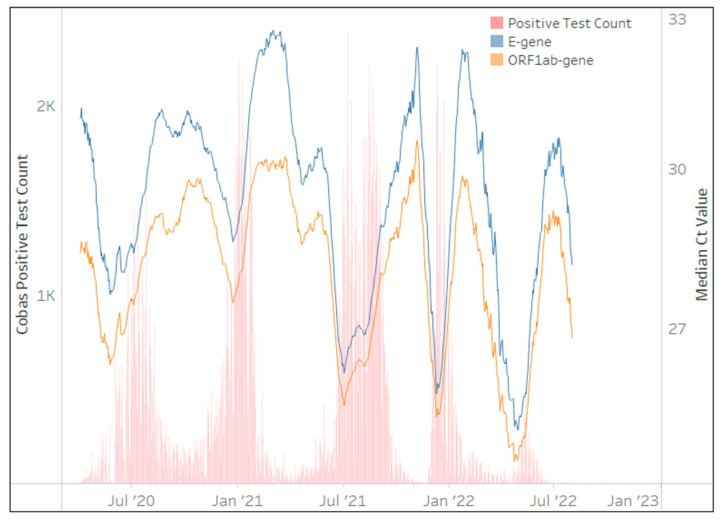
The number of cobas^®^ SARS-CoV-2 tests (daily) reporting the presence of SARS-CoV-2 in red bars between March 2020 and March 2023 (primary vertical axis) overlaid with the median Ct values from each gene target in line plots. The median Ct value of the 28-day centered moving average for each gene target is represented on the secondary vertical axis. The key highlights the assay specific gene targets. Both gene target Ct curves perfectly mirror each other throughout the testing period.

**Figure 7 diagnostics-13-02554-f007:**
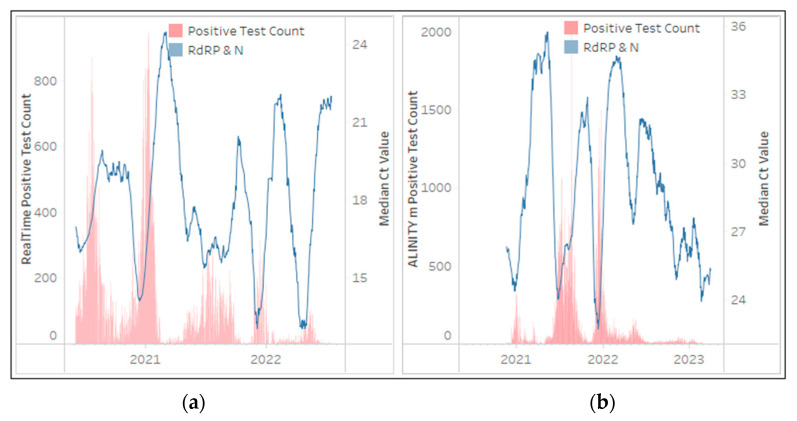
The number of RealTime SARS-CoV-2 (**a**) and ALINITY m SARS-CoV-2 (**b**) tests (daily) reporting the presence of SARS-CoV-2 in red bars between March 2020 and March 2023 (primary vertical axis) overlaid with the median Ct values from each gene target in line plots. The median Ct value of the 28-day centered moving average for each gene target is represented on the secondary vertical axis. The key highlights the assay specific gene targets, which for these assays, both targets are reflected in a single fluorescent channel (visualized as blue) and cannot be differentiated. Testing on the RealTime SARS-CoV-2 discontinued in July 2022.

**Table 1 diagnostics-13-02554-t001:** Description of the six commonly used SARS-CoV-2 open and closed testing platforms within NHLS laboratories.

Assay Name (Manufacturer)	Platform (Type)	Gene Target(s)	Ct Range ^2^
Allplex SARS-CoV-2 (SeeGene Inc, Seoul, Republic of Korea)	BioRad CFX96Touch, Applied Biosystems (open)	*RdRp*, *N*, *E*, *S* ^1^	6–40
TaqPath COVID-19 (Thermo Fisher Scientific, Waltham, MA, USA)	Applied Biosystems (open)	*N*, *Orf1ab*, *S*	5–43
Cobas SARS-CoV-2 (Roche Molecular, Pleasanton, CA, USA)	Cobas 6800/8800 (closed)	*E*, *Orf1ab*	12–42
Xpert Xpress SARS-CoV-2 (Cepheid, Sunnyvale, CA, USA)	GeneXpert (GX) 1, 4, 16 or Infinity-48/80 (closed)	*N2*, *E*	11–45
RealTime SARS-CoV-2 ^3^ (Abbott Molecular, Abbott Park, IL, USA)	*m*2000sp and *m*2000rt (closed)	*RdRp*, *N*	2–31
Alinity m SARS-CoV-2 AMP Kit (Abbott Molecular, Abbott Park, IL, USA) ^3^	Alinity m system (closed)	*RdRp, N*	5–42

^1^ version 2 included *S*-gene targets but in the same fluorescent channel as *RdRp.*
^2^ Ranges as captured through the LIS and aggregated within the CDW environment. These Ct ranges are established by the assay manufacturers, and specific to each assay, and several available in the assay package insert. The LIS reports these ranges as they are drawn from each assays’ LIS interfaced file. ^3^ The two Abbott assays report their gene targets in a single fluorescence channel.

## Data Availability

The National Health Laboratory Service data analyzed in this study is unavailable due to patient privacy and data restrictions. Data resides on the CDW for pathology services. Aggregated data may be accessible through the AARQA (Academic Affairs Research Office and Quality Assurance) division of the NHLS under leadership of KM.

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
