# Peer review of "How South Africa Used National Cycle Threshold (Ct) Values to Continuously Monitor SARS-CoV-2 Laboratory Test Quality"

_diagnostics, 2023, doi:10.3390/diagnostics13152554_

Round 1

Reviewer 1 Report

The authors have analysed SARS-CoV-2 RT-qPCR data collected from diagnostic labs. They have followed the ct value data obtained by several platforms/equipment/kits and propose that it could be used for quality control purposes. The aims of the study should be more focused and stated clearly and it could be confirmed that the study gives the answers. Methods can be improved and should be described more in detail. The interpretation becomes complicated due to the variants of concern and since no comparison of the same samples in different platforms has been done. It would be good to describe how the ct values change when no new VOCs emerge, although the virus occurrence has some seasonality. It is not convincingly described how this approach could be used for quality control. Do the authors suggest that this approach could be used SARS-CoV-2 test only or for other virus tests as well? Below, comments are given for further improvement of the manuscript.

L48 Check the grammar in the sentence (’scale’).

L50 Check if the sentences require commas or periods.

L82 Check the word COVID19 (COVID-19?). Please, correct throughout the text.

L91-93 Usually the aim of the study is written in the last paragraph. Please, add.

L69 All readers are not familiar with LIS laboratory information. Please, describe in more detail.

L101 CDW, please describe. Also, it might help if the authors describe which operator/institute was doing which tasks in the process. (A flow-chart might be considered). 

L108-9 Is it possible to describe the algorithm etc. in more detail? Were the samples analysed every day, also in weekends?

Table 1. The general principles how the ct values were given by the PCR equipment could be described. Were the platforms usually giving results from multiple channels (one fluorescence wavelength used for one target)?

L144 Should the sentence include ’of them’ or ‘of the latter’ or similar?

L152 This kind of writing of numerals (2,095,588 million) is not familiar to me. Please, check, if it is accepted in scientific journals.

L167-70 Is it possible to be more specific? Please, mention when each VOC appeared in South Africa? Is it possible to add a reference?

As the authors state, the possible follow-up samples of one person were not excluded from the data. The possible effect of that could be discussed in the discussion. For how long can one person give a positive nasal swab?

L189 This kind of calculation seems quite misleading, since usually lower ct means more virus gene copies. Would it be possible to use a high ct value (over 50 or something?) instead?

L215-6 Please, define mirroring more accurately (in the methods?). How was it determined, what were the limits?

L277-9 The message of the sentence remained unclear. Please, clarify.

L282-8 Would it be better to move this text to the introduction (aims?) The first paragraph of the discussion seems to give some justifications for the study.

General comment: The study lacks more detailed information, how to use the system in quality control in real life, in practice. Were some signals sent to the laboratories or for platform producers? Can it be real-time or is there always some delay? What level of a CT change would be causing an alarm?

They were included above.

Reviewer 2 Report

The paper entitled: How South Africa used national cycle threshold (Ct) values to continuously monitor SARS-CoV-2 laboratory test quality” gives an overview on how South Africa National Health Laboratory service implemented and managed the SARS-CoV-2 results obtained during COVID pandemic in the country.

The paper gives a good breakdown on the assays in use in South Africa and describes how they monitor SARS-CoV-2 to detect rapidly changes in kit performance, when other quality control measures were less timely. The adding value of this paper, is that highlights a simple method to identify kit mismatch in real time, ensuring a rapid response or switch to other assays if necessary. However, this cannot totally substitute laboratory assessment aiming at evaluating the quality and capacity of the laboratory at higher level.  

Please, consider some major and minor suggestions, which could improve the paper:

Major revisions:

In the method section, it is not reported how you dealt with the different cut off within kits protocols, but in the table 1 it is reported the Ct range as captured in LIS. Four out of six platforms included, reported as superior limit Ct values over 40, achieving 45 as cut off, which correspond to the end of the reaction. How did you managed these values? You reported how you managed the problem for GeneXpert, but how did you deal with the other kits? Considering your aim is to monitor “SARS-CoV-2 laboratory test quality” how did you discriminate false positives and contaminations? What was the mechanism implemented to minimise in your analysis this kind of bias? On the other hand the Abbot system Ct rage is 2-31, which seems too “low” compared with other systems and may suggest an automatic adjustment of the first cycles, so maybe it is not comparable as crude Ct values.  

In the results section, it is reported that gene failure, leading to undetectable amplification, is reported as zero and included in the analysis, lowering considerably the Ct values mean. This must be explained very clearly in the methods as it is confusing, especially after figure 2, where RdRP changed its performance in July 2021 and “surpassed” N gene increasing its Ct value mean due to a lower performance. I would suggest excluding negative results in the calculation, if possible (it depends on the number of samples with drop out). Otherwise, consider to use Ct 45 as negative results, so that in the mean calculation the S gene would show much higher Ct values and conceptually is easier to follow (i.e. both lower performance or drop out would show higher Ct values in all figures being “closer to negative results”).

Introduction:

Despite the objective is clear when reading the whole paper, the research question remains uncertain in the introduction. Reading the paper I think you aimed at describe a very impressive upscale of the laboratory capacity within the country monitoring the quality of the results using Ct values and implementing a CQM system. In the introduction, I would suggest to focus the discussion on the background and the rationale of your research, reporting the risk of a rather sudden and huge upscale of laboratories implementing a new assay in their routine. I would cut all the information on platform in use (including them in the methods section), shortening also the description of the organisation (maybe adding a figure describe the connections and the flow between all laboratories and the NHLS), emphasising what was “the problem” you wanted to investigate and what were the objectives of your research. I would report in the introduction: what are the routine laboratory quality check, the problem of kit shortage, the mismatch of some kits with some variants, the challenge in monitoring kit performance in near real time, the implementation of your network and, finally, the aim of the study with its specific objectives.

Additional suggestions and minor corrections on introduction:

1.       Please, check brackets. The use of many brackets makes difficult to read some sentences. Consider dividing the sentence if the information are relevant or just cut them out.

2.       Lines 58-60: the sentence on “close platforms” is not clear. I think you could consider to rewrite it stressing the problem of shortage and difficulties in procurement.

3.       Lines 64-69: Did you required also the participation to external quality assessment (EQA) schemes? Did you consider specific characteristics of “acceptable” kits (e.g. inclusion of exogenous or endogenous internal controls, approved certification, etc…)? In your assessment, it seems you verified the laboratories quality on-line only, but did you consider running a ring test? exchanging a panel of samples would have help in verifying the kit in use in each laboratory (i.e. checking Limit of detection, cross-reaction and sensitivity/specificity of kits in use) and to compare the Ct values means. If you could not, I would suggest to say it, at it would be a good link also to the first part of the discussion, where you mention EQAs and other quality check.

4.       Line 80: I would say to detect SARS-CoV-2 variants, including VOC (or did you selected possible VOC sequences with typing real time assays?)

5.       Lines 85-88: this sentence is not clear. Did you want to specify each kit had specific viral gene targets and acceptable Ct values, based on LoD?  

Methods:

I would suggest moving part of the introduction in the methods section, such as the kit description. Did the laboratories report also internal control Ct values, despite excluded in the analysis? If not, I would declare it. If you have it, you could add a sentence reporting whether all reactions had IC Ct values in acceptable range and how you dealt with unacceptable values (if any).

Line 101-102: From this sentence, it seems all recorded data included Ct values, but in the results section (line 144) it seems only 95.2% of test reported actually Ct values for their gene target. Can you clarify?

Line 111: here you reported these 6 platforms covered 93% of test performed, but in figure 1 the proportion is 94.2%. Please, use the same proportion.

Results:

Line 146-148: this sentence is not clear. In which sense low Ct values are followed by an increased SARS-CoV-2 positivity? Regardless the Ct value, if the run and samples are validated and only positive samples are reported (despite I personally doubt the real positivity of samples with Ct values at 45), how is the positivity affected by the low of high values of Ct? Looking at the graph, I guess you wanted associate the low Ct value peak with a consequential increase in the number of positive test count in terms of epidemiological curve. If that is the case, please rephrase and clarify how you associated low Ct values to an increase number of positive tests.  

Discussion:

Lines 299-305: it seems that analysing big numbers can minimise the impact of other variables. I would suggest adding to this sentence, that in any case it is necessary to implement quality control at laboratory level (EQA or inter-laboratory exercises) to assess the intrinsic laboratory quality and capacity. On the other hand, I agree the use of a large set of data helps in observe a general phenomenon and is very valuable when used to detect kit failures or changing performances.

Overall the paper reads very well, but some sentences could benefit a rearrangement to clarify their meaning. 

Reviewer 3 Report

Dear Authors,

Your present article has been reviewed,

This manuscript deserves attention since it highlights an important topic related the usage of Ct values in the monitoring of SARS-CoV-2 laboratory test quality in South Africa.

The article contains many weak points regarding the English language, the Presentation of data and a conclusion of the study.

Kindly find below a list of my comments and suggestions regarding this work:

01- The Abstract section is very weak, its anatomy is not clear (No purpose, no Materials and Methods, No Results neither a Conclusion in the Abstract)

02- The first sentence in the Abstract, Lines 19-20, Authors are invited to paraphrase the sentence since it is very weak.

03- Concerning the List of Keywords, Authors are invited to remove some of these terms, (5 keywords are good enough).

04- In the Introduction section, Line 43, at the end of this line, Authors are invited to put a reference for this idea.

05- In the Introduction section, Page 2, Third line, the Reference "13" is not related to the sentence!! Authors are invited to remove it from this sentence.

06- In The Results section, Page 5, When authors talk about "S-gene target failure (SGTF), authors are invited to cite the following reference since it talk about this variations using the TaqPath kit.

Ref: The emergence of SARS-CoV-2 variant (s) and its impact on the prevalence of COVID-19 cases in the Nabatieh Region, Lebanon

07- In the Discussion section, Page 10, First line, Authors are invited to cite the following reference:

Ref: Risk Markers of COVID-19, a Study from South-Lebanon

08- Authors are invited to put a Conclusion section in this article.

Best Regards,

The present article needs English polishing by native English speaker, for example the first sentence in the Abstract is not clear for readers.

Best Regards,

Round 2

Reviewer 1 Report

The authors have responded to the comments and made improvements, which can be accepted.

Reviewer 3 Report

Dear Authors,

I would like to thank you for the correction you made,

The article is way better in its present form for publication,

Best Regards,